# The Bacterial Microbiota of Artisanal Cheeses from the Northern Caucasus

Tatiana V. Kochetkova [1,*], Ilya P. Grabarnik [2], Alexandra A. Klyukina [1], Kseniya S. Zayulina [1], Liliya A. Gavirova [3], Polina A. Shcherbakova [3], Gennady S. Kachmazov [4], Andrey I. Shestakov [3], Ilya V. Kublanov [1] and Alexander G. Elcheninov [1]

[1]  Winogradsky Institute of Microbiology, Federal Research Center of Biotechnology of the Russian Academy of Sciences, Moscow 117312, Russia; alexandra.a.popova@gmail.com (A.A.K.); zauylinakc@gmail.com (K.S.Z.); kublanov.ilya@gmail.com (I.V.K.); elcheninov.ag@gmail.com (A.G.E.)
[2]  Applied Genomics Laboratory, SCAMT Institute, ITMO University, Saint Petersburg 197101, Russia; ilyagrabh@gmail.com
[3]  Faculty of Biology, Lomonosov Moscow State University, Moscow 119234, Russia; gavirovaliliya@gmail.com (L.A.G.); shcherbakovapa@gmail.com (P.A.S.); 6.ok.off@mail.ru (A.I.S.)
[4]  Department of Food Technology, Faculty of Chemistry, North Ossetian State University after K.L. Khetagurov, Vladikavkaz 362025, Russia; kgssogutfi@yandex.ru
*  Correspondence: kochetkova.tatiana.v@gmail.com

**Abstract:** In this study, we used culture-independent analysis based on 16S rRNA gene amplicons and metagenomics to explore in depth the microbial communities and their metabolic capabilities of artisanal brine cheeses made in the North Caucasus. Additionally, analysis of organic acid profiles was carried out for cheese characterization. Twelve cheese samples (designated as 05SR–16SR) from various artisanal producers were taken from five different villages located in Northern Ossetia–Alania (Russia). These cheeses were made using methods based on cultural traditions inherited from previous generations and prepared using a relatively uncontrolled fermentation process. The microbial diversity of Caucasus artisanal cheeses was studied for the first time. The results showed a diverse composition in all cheeses, with *Bacillota* (synonym *Firmicutes*) (9.1–99.3%) or *Pseudomonadota* (synonym *Proteobacteria*) (0.2–89.2%) prevalence. The microbial communities of the majority of the studied cheeses were dominated by lactic acid bacteria (LAB) genera, like *Lactococcus* (10.3–77.1% in 07SR, 09SR, 10SR, 11SR, 13SR, 15SR, 16SR), *Lactobacillus* (54.6% in 09SR), *Streptococcus* (13.9–93.9% in 11SR, 13SR, 14SR, 15SR), *Lactiplantibacillus* (13.4–30.6% in 16SR and 07SR) and *Lentilactobacillus* (5.9–14.2% in 09SR, 10SR and 13SR). Halophilic lactic acid bacteria belonging to the *Tetragenococcus* genus accounted for 7.9–18.6% in 05SR and 06SR microbiomes. A distinctive feature of Ossetia cheese microbiomes was the large variety of halophilic proteobacteria, and in some cheeses they prevailed, e.g., *Chromohalobacter* (63–76.5% in 05SR and 06SR), *Psychrobacter* (10–47.1% in 08SR, 11SR, 12SR), *Halomonas* (2.9–33.5% in 06SR, 08SR, 11SR and 12SR), *Marinobacter* (41.9% in 12SR) or *Idiomarina* (2.9–14.4% in 06SR, 08SR and 11SR samples). Analysis of the genomes assembled from metagenomes of three cheeses with different bacterial composition revealed the presence of genes encoding a variety of enzymes, involved in milk sugar, proteins and lipid metabolism in genomes affiliated with LAB, as well as genes responsible for beneficial bioamine and bacteriocin synthesis. Also, most of the LAB did not contain antibiotic resistance genes, which makes them potential probiotics, so highly demanded nowadays. Analysis of the genomes related to halophilic proteobacteria revealed that they are not involved in milk fermentation; however, the search for "useful" genes responsible for the synthesis of beneficial products/metabolites was partially positive. In addition, it has been shown that some halophiles may be involved in the synthesis of inappropriate bioactive components. The results obtained by culture-independent analyses confirm the importance of using such techniques both to clarify the quality and health-promoting properties of the product, and to look for probiotic strains with specified unique properties. This study has shown that traditional dairy foods may be a source of such beneficial strains.

**Keywords:** brine cheese; NGS; traditional dairy products; microbial community; lactic acid bacteria; halophilic bacteria; the Caucasus; metagenomics

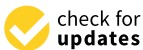



## 1. Introduction

Cheese has always been valued by men both as an everyday food and as a gourmet dish. Cheese is mentioned as a major component of the daily diet of the Babylonian dynasty. In the Roman Empire, cheese was an integral part of the feasts of the patricians. In ancient Greece, cheese making was as well-known as in our age. It is proved by Aristotle's treatise, where the processes of milk coagulation and cheese making technique are described. Nowadays cheese is a traditional product widely consumed around the world. It is high in protein (up to 25%), milk fat (up to 60%) and minerals (up to 3.5%, not including table salt). Proteins in cheese are better assimilated than proteins in other dairy products. It is also consumed by people with lactose intolerance because some amount of the lactose present in milk is hydrolyzed during fermentation, and the rest is processed into whey during the production of cheese [1].

Traditional products (including cheeses) are frequently consumed, usually passed down from one generation to the next, are prepared precisely defined according to its gastronomic heritage, with little or no processing/manipulation and are associated with a particular locality, region or country [2]. It is not a secret that the microbiota common to these foods are linked to health-promoting and beneficial properties for humans, such as up-regulation of the immune system, strengthening the intestinal–brain barrier, minimization of carcinogenic effects and more [3]. It was revealed that lactic acid bacteria species, isolated from cheeses, could be producers of such bioactive molecules or enzymes as peptidases and lipases [4], vitamins [3], GABA (γ-aminobutyric acid) [5], antibacterial peptides and peptides inhibiting the increase of blood pressure [6]. Microorganisms isolated from artisanal cheeses secrete metabolites with anti-cancer effects [7–9], exhibit antimicrobial activity against pathogenic bacteria [10], participate in lowering cholesterol levels [11], enhance the immune system by increasing the level of IgA and T helper cells [7], etc. On the other hand, we must not forget about foodborne infection and foodborne intoxication, caused by "unwanted" microbiota, which can occur for various reasons (disruption or low hygienic level of the production process or transportation, storage conditions and so on) in traditional dairy foods, including cheeses [12,13].

Ultimately, the identification of microbial communities that contribute significantly to the quality, safety, beneficial properties and flavor characteristics of cheese and their potential functional properties is of high importance. The cheese microbiome is a dynamic community that can change throughout ripening, and factors such as raw materials, aging, production process, storage conditions and specific product characteristics affect its diversity. The next-generation sequencing (NGS) technologies have significantly enhanced the capability to characterize and describe the cheese microbiome compared to commonly used culture-dependent methods. A lot of work on microbiota of commercial and artisanal cheeses has already been carried out all over the world using NGS (see reviews [14,15]). Studies devoted to microbiome analysis of cheese, produced in Russia, with its huge diversity of ethnic groups and, therefore, their own traditions in preparing dairy products, remains extremely scarce [16–19].

The Caucasus is a geographic mountainous region located on the border between Europe and Asia, covering the territory of Russia, Georgia, Azerbaijan and Armenia. This territory has long been famous for its ancient traditions in the preparation of dairy products. A distinctive feature of the Caucasian cuisine is the frequent and abundant inclusion of cheeses. Exclusively brine-type cheeses are in usage. The region of North Ossetia–Alania, located in the North Caucasus, occupies an area of 8000 km$^2$ and is characterized by a moderate continental climate and a long growing season, which allows obtaining high-quality milk from farm animals. The climatic zone of North Ossetia, softened by the proximity of the mountains, with mild winters and long rainy summers can affect the ripening of cheese, as this process proceeds at environmental conditions. Traditional cheese in North Ossetia is made by adding rennet extract (prepared from veal's stomach chopped into pieces and steeped for 2–3 days in acidified cheese whey) to raw milk (sheep's, goat's or cow's) and incubation of this mix at room temperature. After full coagulation, the cheese

mass is left to stand in the tank to remove the whey, and after the formed cheese is dry, salted by rubbing the surface (young soft cheese). For longer storage, cheese is consistently salted and dried, repeating these procedures, and then placed in a concentrated brine solution (approximately 1 kg of salt per liter of water). In this form the cheese can mature for 3–12 months (semi-hard cheese) and even up to 1–1.5 years (hard cheese). Studies on the microbial composition of such traditional fermented foods have never been documented.

The aim of this work is to explore the prokaryote diversity of home-made authentic Ossetia cheeses, as well as to investigate the gene sets related with metabolic activities and probiotic functions that could be involved in their sensory profile, safety and beneficial properties. In this study we used culture-independent NGS analysis based on 16S rRNA gene amplicons of bacterial communities of 12 traditional cheeses and metagenomics of three of them.

## 2. Materials and Methods

### 2.1. Sample Collection and DNA Extraction

Cheeses prepared from cow's milk by traditional methods were sampled in local markets and houses in villages and towns of various districts of North Ossetia–Alania in spring 2022 (Table 1). Cheese wheels were transported to the laboratory at 4 °C. For DNA fixation, 2 mL of product (a 20 mL syringe with the front end cut off was used to sample 20 mL of cheese, a cylinder was cut out, capturing both the outside and the inside of the piece of cheese, then the cylinder was pressed through a metal net with a hole size of $0.8 \times 0.5$ mm, 2 mL was taken from this sample) were mixed with 2 mL of fixing buffer (100 mM EDTA, 100 mM Tris-HCl, 150 mM NaCl; pH 8.2). DNA extraction and all other manipulations were carried out within 7 days after sampling. Fixed product samples were centrifuged at $18,000 \times g$ for 20 min. Total DNA was extracted from the pellets using DNeasy PowerLyzer Microbial Kit (Qiagen, Hilden, Germany) and FastPrep-24™ 5G grinder (MP Bio, Irvine, CA, USA) according to the manufacturer's instructions. DNA concentrations were detected by Qubit 2.0 (Invitrogen/Life Technologies, Vallejo, CA, USA). DNA was stored at $-20$°C.

**Table 1.** Samples and types of artisanal cheeses, their location, pH and DNA analysis.

| Sample Designation | Type of Cheese | Time of Ripening in Salt Brine | District | GPS | pH | DNA, ng/µL | DNA Analysis |
|---|---|---|---|---|---|---|---|
| 05SR | SH | 6 months | Irafsky | 42.936421, 43.818312 | 5.0 | 38.6 | V4 16S |
| 06SR | SH | 12 months | Irafsky | 42.906694, 43.857689 | 5.3 | 104.3 | V4 16S Metagenomics |
| 07SR | SH | 6 months | Irafsky | 42.906694, 43.857689 | 5.0 | 92.1 | V4 16S Metagenomics |
| 08SR | SH | ND | Irafsky | 42.906694, 43.857689 | 5.0 | 9.9 | V4 16S |
| 09SR | H | ND, but long dried on a shelf | Irafsky | 42.906694, 43.857689 | 4.5 | 62.0 | V4 16S |
| 10SR | S | Without ripening | Alagirsky | 42.674195 43.909169 | 5.5 | 56 | V4 16S |
| 11SR | SH | 12 months | Prigorodny | 42.965969, 44.773623 | 5.3 | 62 | V4 16S |
| 12SR | SH | 12 months | Prigorodny | 42.965969, 44.773623 | 5.3 | 97.6 | V4 16S |

**Table 1.** *Cont.*

| Sample Designation | Type of Cheese | Time of Ripening in Salt Brine | District | GPS | pH | DNA, ng/μL | DNA Analysis |
|---|---|---|---|---|---|---|---|
| 13SR | S | Without ripening | Ardonsky | 43.081601, 44.424295 | 5.0 | 38 | V4 16S Metage-nomics |
| 14SR | SH | 8 months | Alagirsky | 42.674195 43.909169 | 5.5 | 21.6 | V4 16S |
| 15SR | S | Without ripening | Alagirsky | 42.674195 43.909169 | 5.0 | 3.4 | V4 16S |
| 16SR | SH | 1 month | Alagirsky | 42.674195 43.909169 | 5.3 | 6.8 | V4 16S |

SH—semi-hard cheese, H—hard cheese, S—soft cheese, ND—no data, V4 16S—NGS analysis of 16S rRNA gene amplicons of bacterial communities.

## 2.2. Organic Acids Analysis

Sample preparation and organic acids analysis were performed as described previously [19]. The pH value of the samples was measured using pH indicator strips (pH-fix 0.0–6.0, Macherey-Nagel, Dueren, Germany).

## 2.3. Library Preparation and Sequencing

For amplicon-based library preparation the V4 hyper-variable region of the 16S rRNA gene was amplified with a pair of primers 515F (5′-GTGBCAGCMGCCGCGGTAA-3′) [20] and Pro-mod-805R (5′-GGACTACHVGGGTWTCTAAT-3′) [21], and then adaptors and dual indices were added in a secondary amplification as described previously [22]. The libraries were sequenced using the MiSeq system (Illumina, San Diego, CA, USA). The libraries were prepared and sequenced in two technical replicates for each sample.

Preparation of DNA libraries for shotgun metagenomic sequencing was carried out using KAPA HyperPlus kit (KAPA, Wilmington, MA, USA) according to manufacturer recommendations. The manipulations include enzymatic fragmentation of DNA which resulted in fragments with length of 500–700 bp, ends polishing and A-tailing, ligation of specific adapters for sequencing (Nextera Mate Pair Library Prep Kit, Illumina, San Diego, CA, USA) as well as amplification of obtained libraries. Metagenomic sequencing was performed using Illumina NovaSeq 6000 platform (Illumina, San Diego, CA, USA).

## 2.4. Bioinformatics and Statistical Analysis

For amplicon sequences adapter trimming and demultiplexing were performed as described earlier [23]. The obtained reads were filtered and processed using dada2 package v.1.14.1 [24] (parameters: truncLen = 220, maxN = 0, maxEE = 2, truncQ = 2) resulting in identification of the amplicon sequence variants (ASV). Taxonomic assignment of ASV was performed using dada2 package v.1.14.1 with native Bayesian classifier [25] and Silva 138.1 database [26]. Biodiversity indexes such as Shannon [27], InvSimpson [28] and Chao1 [29] indexes were calculated using the phyloseq v.1.3 package [30]. To estimate the dissimilarity of the microbial composition of cheeses (i.e., the beta-diversity), a non-metric multidimensional scaling (NMDS) was performed as an ordination method based on ASV summarized table and the Bray–Curtis dissimilarity indices using phyloseq and vegan v.2.6.-4 packages (accessed on 23 June 2023 https://CRAN.R-project.org/package=vegan). Visualization of the results was performed with the ggplot2 package (accessed on 23 June 2023 https://ggplot2.tidyverse.org).

## 2.5. Metagenome Assembly and Dominant Genomes Reconstruction

Raw sequences of three cheese samples' (06SR, 07SR, 13SR) metagenomes were filtered using trim tool (quality limit = 0.03, maximum ambiguous nucleotides = 2, and minimum length = 80) in CLC Genomic Workbench v.10 software (Qiagen). Prepared

high-quality reads were assembled using the metaSPAdes v3.15.5 [31]. Contigs with length less than 500 bp were eliminated from the assemblies. To generate metagenome-assembled genomes (MAGs) from metagenomes, and assembled contigs were processed using the metaWRAP v1.3.2 standard pipeline [32]. The metaWRAP::Binning module was used to reconstruct bins from contigs and mapped reads by three independent binning tools (MaxBin2 [33], metaBAT2 [34] and CONCOCT [35]) with default settings. In the metaWRAP::Bin_refinement module multiple combinations of obtained bins were compared, filtered by completeness (-c 80; more than 80%) and contamination (-x 10; less than 10%) estimation with CheckM v1.1.6 [36] and consolidated to the refined bin set. Abundance and statistical calculations were performed in the metaWRAP::Quant_bins module by Salmon v1.10.1 [37]. Taxonomic classification of MAGs was performed using the GTDB-Tk v2.1.1 [38] with GTDB_r207 database as reference.

### 2.6. Gene Prediction and Functional Annotation

Functional analysis was performed using a set of MAGs from three cheese samples. Gene calling was performed using Prodigal v.2.6.3 [39] and Prokka v1.12 [40] was used for rapid genome-wide annotation.

Carbohydrate-active enzymes (CAZymes) genes were searched in MAGs using db-CAN v.4 [41] with hmmer tool [42]. To found putative proteases blast search was performed using in silico translated MAGs as queries and MEROPS_scan [43] with an e-value threshold of $1 \times 10^{-5}$; positive hits were analyzed with SignalP v.6.0 [44] to identify extracellular enzymes. Putative esterase was identified in translated MAGs using blast search against a database of bacterial lipolytic enzymes [45] with an e-value threshold of $1 \times 10^{-10}$ and minimal coverage of 30%.

Antimicrobial resistance genes in MAGs were determined using the ABRicate tool (accessed on 20 June 2023 https://github.com/tseemann/abricate). Each sample bin set was screened using the CARD database [46] with compared regions' minimal identity of 80% and minimal coverage of 90%. Feature annotation of MAG sequences produced by Prokka and genomes in fasta format were used for biosynthetic gene clusters (BGC) identification by antiSMASH bacterial version 7.0 [47]. The search for bacteriocins among secondary metabolites identified by antiSMASH was carried out by selecting gene clusters containing genes with the corresponding annotation in the Pfam and TIGRFam databases and their further annotation using NCBI BlastP analysis.

Genes encoding enzymes involved in biogenic amines and GABA production were searched using blast with characterized enzymes as queries and translated MAGs as databases (e-value of threshold $1 \times 10^{-5}$). Functions of positive hits were checked with blast against the SwissProt database [48].

### 2.7. Data Availability

All sequencing data used for 16S rRNA amplicon analysis were deposited into the NCBI SRA database under BioProject number PRJNA789261 (Table S1). Metagenomic sequences are available in the Genbank database under accession numbers JARIFJ000000000, JARIFN000000000 and JARIFP000000000 within Bioproject PRJNA907749.

## 3. Results

We analyzed 12 samples of home-made cheeses from different districts of North Ossetia-Alania (Table 1, Figure 1).

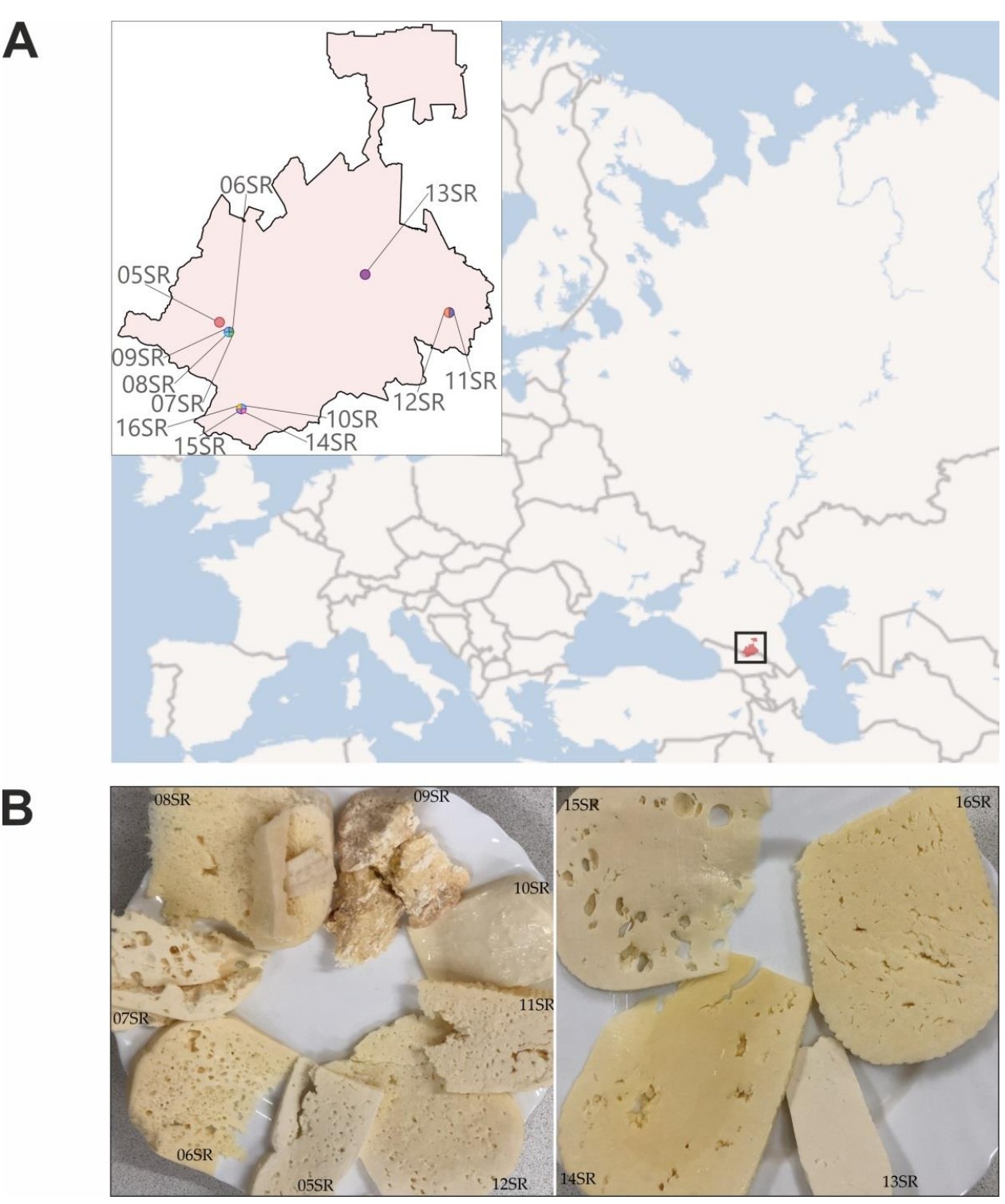

**Figure 1.** Geographic location of sampling sites (**A**) and appearance of sampled cheeses (**B**) with designations.

### 3.1. Organic Acids Content

The profile of organic acids (OA), contained in the cheeses, include formate, acetate, propionate, n-butyrate, lactate, succinate, malate and citrate. The major organic acid in all samples was lactate (Figure 2) with concentration varying from 11.7 (in 09SR sample) to 37 mM (in 05SR). Acetate was detected in all cheeses, except sample 09SR, with concentration from 1.2 mM (for 12 SR) to 19.1 (for 10SR). Other volatile fatty acids like formate, propionate and n-butyrate were identified for all samples in negligible concentrations from

0.3 to 1.1 mM, except sample 05SR, which contained 3 mM of propionate. Succinate was measured for all samples and its concentration varied from 1.4 mM (for 05SR, 07SR and 13SR samples) to 3.4 mM (10SR), as well as citrate, its concentration varied from 0.1 mM (09SR) to 3.3 mM (13SR). Small amounts of malate were detected in six samples from 0.2 mM (12SR and 15SR) to 1 mM (09SR). The total concentration of all OA varied greatly among the samples—from 15.9 mM (for 09SR) to 56.3 (10SR). pH values of cheeses were in a narrow range from 4.5 to 5.5.

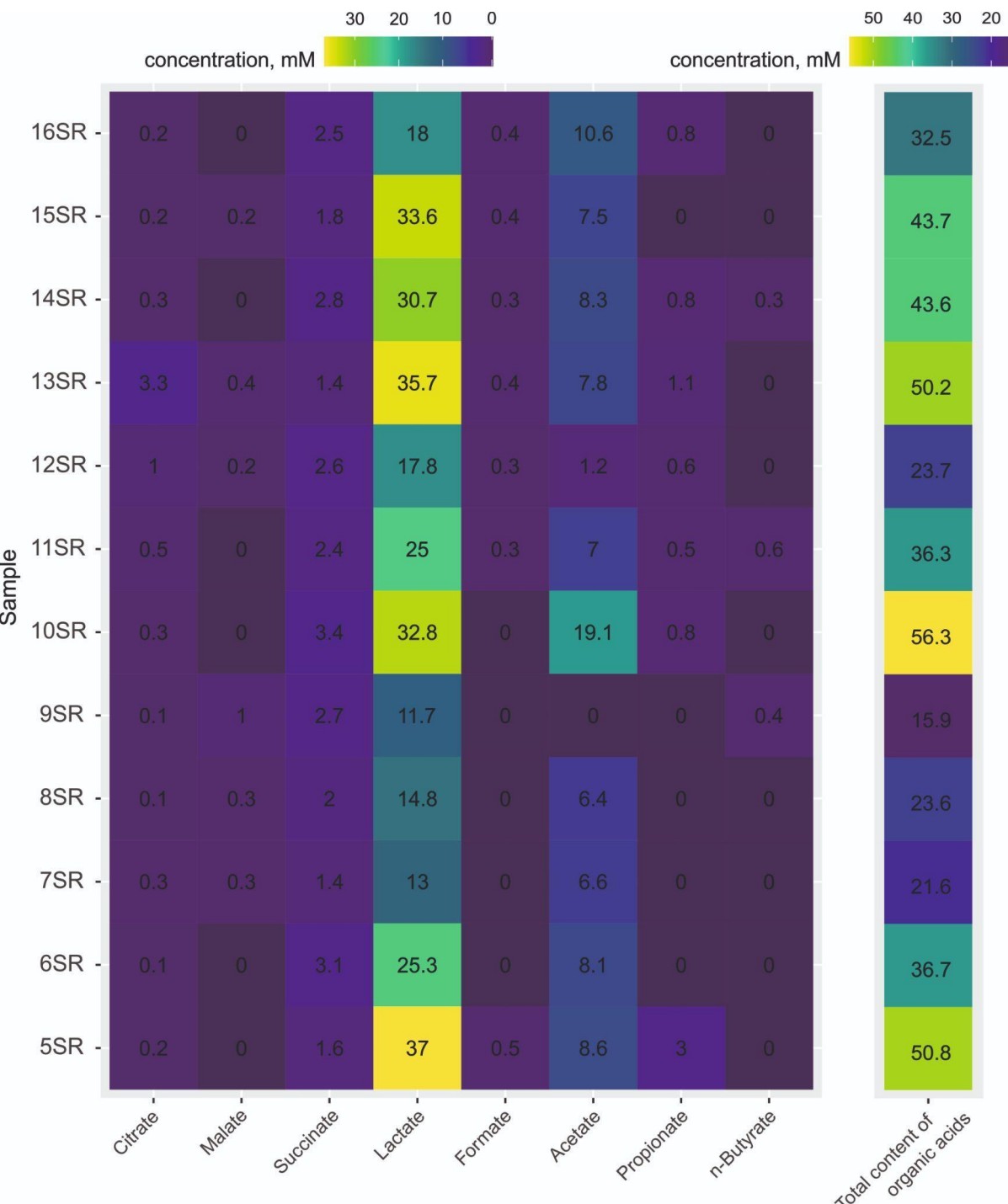

**Figure 2.** The organic acids content in cheeses.

### 3.2. Sequencing, Taxonomic Annotation and Genome Reconstruction

3.2.1. Taxonomic Analysis by Amplicons of V4 Region of the 16S rRNA Gene Sequencing

A total of 279,232 reads with an average length of 250 bp were obtained from sequencing of 12 samples. After filtering, denoising and chimera detection 179,320 reads were retained representing 288 unique sequences. The obtained ASV were assigned to more than one hundred genera (145 unique genera + 37 NA) within 25 phyla, but 98.9% of the total number of sequences were affiliated to *Firmicutes* and *Proteobacteria* (synonymes *Bacillota* and *Pseudomonadota*, respectively, according to GTDB database, accessed on 29 June 2023 https://gtdb.ecogenomic.org/) (Figure 3).

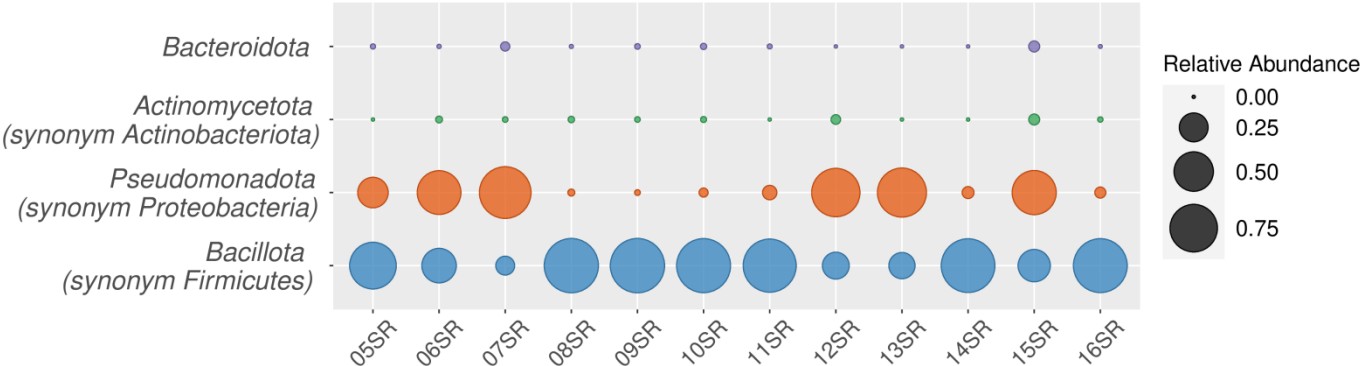

**Figure 3.** Composition of cheeses' microbial communities at the phylum level. Only phyla containing genera, which made up ≥0.5% of the total microbial community for at least one sample, are shown.

To estimate overall diversity in all samples analyzed the alpha-diversity indexes were calculated for each sample. According to the Shannon Index, cheeses 11SR and 08SR possessed the highest biodiversity in comparison with other studied products (Figure 4A). The maximum value of 2.80 was observed for the 11SR sample. Indeed, this cheese had the highest microbial composition and was presented by *Streptococcus* (14% of the total community), *Lactococcus* (10%), *Idiomarina* (14.4%), *Halomonas* (16%), *Psychrobacter* (15%), *Marinilactibacillus* (5.5%), *Cobetia* (5.3%), *Chromohalobacter* (3.1%), *Pseudoalteromonas* (4%), *Staphylococcus* (3.5%), *Lentilactobacillus* (0.9%), *Lactobacillus* (0.8%), *Salinicola* (0.7%) and *Enterococcus* (0.5%) species (Figure 5). The sample 08SR had very similar diversity, however in other proportions. Samples 05SR-07SR, 09SR, 14SR-16SR had lower biodiversity levels with 0.45–1.31 values of Shannon index. The dominant genera in these samples were: *Chromohalobacter* (05SR, 06SR), *Lactococcus* (07SR, 09SR, 16SR), *Lactobacillus* (09SR), *Streptococcus* (14SR, 15SR), *Lactiplantibacillus* (07SR) and *Tetragenococcus* (06SR). The middle index values were observed for 10SR (2.24), 12SR (2.15) and 13SR (1.76) samples. The microbial composition of the sample 05SR was relative to 06SR — dominance of *Chromohalobacter* species (77%), followed by *Tetragenococcus* (8%), *Staphylococcus* (6%), *Streptococcus* (1.4%), *Weissella* (3.6%) and *Lactobacillus* (0.9%). *Streptococcus* was the dominant genus in cheeses 13SR (53%), 14SR (94%) and 15SR (83%). The community of the sample 16SR consisted of only *Lactococcus* (77%), *Lactiplantibacillus* (13.4%) and *Streptococcus* (0.2%) representatives. The analysis of the diversity based on Chao1 and the inverse Simpson indexes revealed relatively similar trends. The Chao1 indexes varied from 23 (06SR) to 89 (08SR), while inverse Simpson values—from 1.18 (14SR) to 11.67 (11SR).

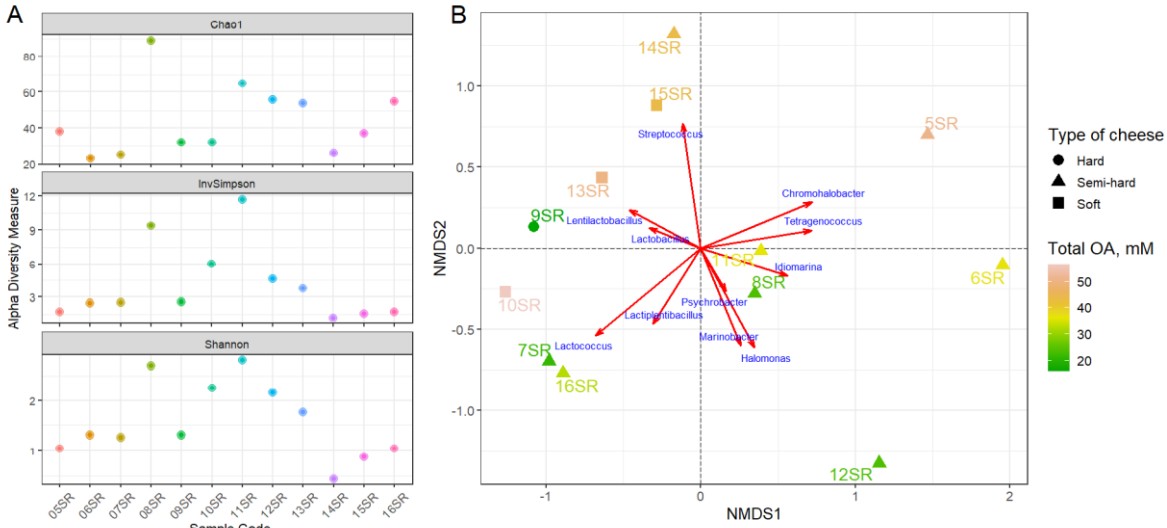

**Figure 4.** Alpha- (**A**) and beta-diversity (**B**) metrics of analyzed cheeses. Alpha-diversity presented with the values of the Chao1, Shannon and inverse Simpson diversity indices. Cheese's microbiomes dissimilarity displayed by NMDS plots (stress value = 0.0906). Color grading corresponds to the concentration of total organic acids (OA) for each product; the shape of points represents the time of cheese ripening in salt brine. Red vectors indicate the correlation of the dominant genera abundance with the axes of ordination and their statistical significance based on a permutation test (2000 permutations, *p*-value < 0.05), i.e., demonstrate the strength and direction (within the present coordinates) of influence of certain bacterial taxa on sample clustering.

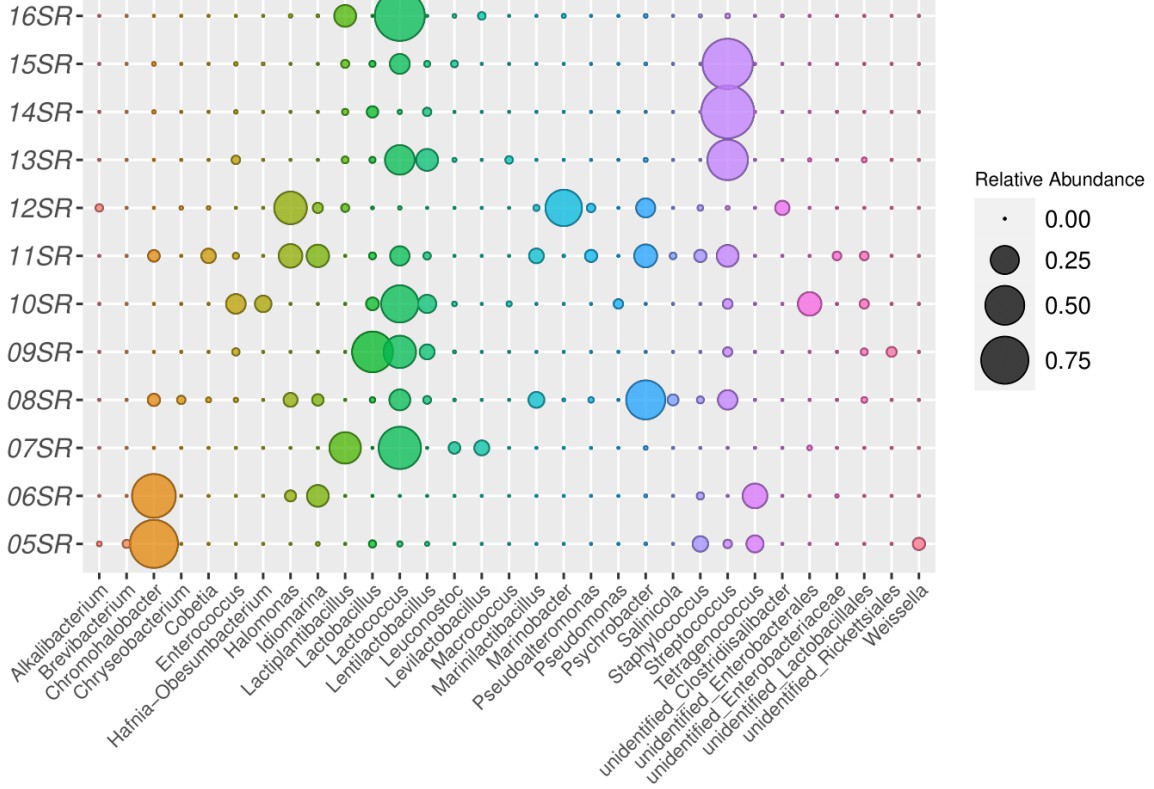

**Figure 5.** Taxonomic assignment at the genus level for the 50 most abundant ASVs found in microbiomes of studied cheeses.

Within the non-dominant population determined by read abundance of 16S rRNA gene marker, bacteria related to food sanitary quality of less than 3.3% of total richness, were identified: *Escherichia-Shigella* (less than 0.5%), *Klebsiella* (less than 0.9%), *Acinetobacter* (less than 3.3%), *Raoultella* (0.2%), *Chryseobacterium* (1.4%), *Rothia* (0.3%), *Ralstonia* (0.1%), etc. There were no sequences with annotation related to pathogenic bacteria such as *Salmonella*, *Listeria*, *Yersinia*, *Mycobacterium*, etc. (Table S2).

The comparison of the microbiomes of different cheeses was performed with NMDS using Bray–Curtis dissimilarity indices based on an ASV table. The microbial components of each sample (ASV) determine the spatial distribution of points, i.e., the biodiversity of the products. The cheese samples were evenly distributed across the space of the NMDS plot (Figure 4B), which may indicate a wide diversity of microbiome composition. However, reflection of the metadata on the points makes it possible to evaluate the general patterns of sample characteristics depending on the similarity of the microbial community: cheeses with a low concentration of organic acids (06SR, 07SR, 08SR, 11SR, 12SR, 16SR) are located in the central and lower parts of the graph, where *Lactococcus*, *Halomonas*, *Idiomarina*, *Marinobacter* and *Lactiplantibacillus* domination was observed. The same group covers almost the entire set of semi-hard type cheeses, except for 05SR and 14SR. The opposite group of samples in the upper half of the plot (05SR, 13SR, 14SR, 15SR) has a high concentration of organic acids, includes an equal amount of soft and semi-hard type cheeses and has *Streptococcus*, *Lentilactobacillus*, *Chromohalobacter* and *Tetragenococcus* as the dominant bacteria. Samples 09SR and 10SR did not fit into groups formed on the plot with their characteristics. The small distance between the points may indicate a significant similarity of samples 07SR and 16SR, as well as 08SR and 11SR. The distance between samples 05SR, 06SR and 12SSR, which have different compositions of the microbiomes with the dominance of *Chromohalobacter*, *Tetragenococcus*, *Halomonas* and *Idiomarina*, was also noticeable.

### 3.2.2. Metagenomic Assembly, Gene Prediction and Functional Annotation

Metagenomes of three cheese samples 06SR, 07SR and 13SR were used for assembling and reconstructing bacterial MAGs. The metagenomes assembly sizes for samples were 26.53, 26.16 and 45.75 Mb respectively as shown in Table 2. Mapping reads back to contigs (with length $\geq$ 500 bp), 98.72%, 96.83% and 68.15% of the total reads were recruited with mean coverage $423\times$, $300\times$ and $114\times$ respectively. After metaWRAP binning module with CONCOCT, MetaBAT2 and MaxBin2 usage, the total number of primary 203 bins were generated, number of which decreased after filtering by completeness and contamination level and refinement algorithm to final 21 MAGs (5, 6 and 10, respectively). Characteristics of MAGs as size (1.52 to 6.06 Mb), completeness and contamination percentage (average 95.98% and 1%, respectively), GC content, number of scaffolds and N50 are summarized in Table 3.

**Table 2.** General features of metagenomic assemblies.

| Sample | Size (Mb) | Scaffolds | N50 | Mapped Reads | Average Mean Coverage | Assembly ID |
|---|---|---|---|---|---|---|
| 13SR | 45.75 | 25125 | 4958 | 98.72% | 423 | GCA_029266255.1 |
| 06SR | 26.53 | 12922 | 7578 | 96.83% | 300 | GCA_029255955.1 |
| 07SR | 26.16 | 7794 | 15908 | 68.15% | 114 | GCA_029255935.1 |

**Table 3.** Metagenome-assembled genomes features and classification.

| Sample | Bin # | Compl., % | Contamin., % | GC, % | Size, Mb | Scaffolds | N50 | Species |
|--------|-------|-----------|--------------|-------|----------|-----------|------|---------|
| 06SR | bin.5 | 100 | 0.336 | 47 | 2.73 | 32 | 169731 | *Idiomarina* sp. |
| | bin.3 | 99.93 | 0 | 60.4 | 2.77 | 52 | 83372 | *Halomonas* sp. |
| | bin.2 | 98.99 | 0.862 | 60.9 | 3.37 | 83 | 85018 | *Chromohalobacter japonicus* |
| | bin.4 | 93.77 | 0.883 | 35.6 | 2.25 | 182 | 17110 | *Tetragenococcus halophilus* |
| | bin.1 | 92.73 | 2.55 | 32.9 | 2.65 | 272 | 14553 | *Staphylococcus equorum* |
| 07SR | bin.2 | 100 | 0.793 | 37.6 | 2.17 | 92 | 77980 | *Leuconostoc mesenteroides* |
| | bin.3 | 99.62 | 0.377 | 34.9 | 2.32 | 73 | 45232 | *Lactococcus lactis* |
| | bin.4 | 99.06 | 0 | 45.5 | 2.61 | 154 | 62186 | *Levilactobacillus brevis* |
| | bin.5 | 98.57 | 1.075 | 58.2 | 6.06 | 565 | 17370 | *Pseudomonas_E helleri* |
| | bin.6 | 96.29 | 2.777 | 44.6 | 3.08 | 97 | 55608 | *Lactiplantibacillus plantarum* |
| | bin.1 | 86.92 | 2.064 | 55.8 | 4.04 | 720 | 6846 | *Enterobacter hormaechei_A* |
| 13SR | bin.2 | 99.25 | 0 | 41.8 | 2.28 | 80 | 36800 | *Lentilactobacillus kefiri* |
| | bin.5 | 99.11 | 0.034 | 34.8 | 2.26 | 217 | 19019 | *Lactococcus lactis* |
| | bin.3 | 98.76 | 2.777 | 45 | 2.85 | 168 | 25798 | *Lactiplantibacillus plantarum* |
| | bin.10 | 98.16 | 0.478 | 38.9 | 1.97 | 93 | 35113 | *Streptococcus thermophilus* |
| | bin.6 | 96.17 | 0.187 | 39.8 | 1.85 | 78 | 38196 | *Streptococcus parasuis* |
| | bin.7 | 94.8 | 3.37 | 37.6 | 2.8 | 200 | 19197 | *Enterococcus faecalis* |
| | bin.4 | 94.27 | 0.552 | 37.2 | 1.83 | 145 | 18105 | *Macrococcus caseolyticus* |
| | bin.8 | 93.26 | 0.084 | 37.5 | 1.6 | 79 | 32698 | *Streptococcus macedonicus* |
| | bin.1 | 91.48 | 0.754 | 39.4 | 1.78 | 186 | 12108 | *Streptococcus dysgalactiae* |
| | bin.9 | 84.43 | 1.131 | 38.7 | 1.52 | 131 | 14625 | *Pediococcus parvulus* |

After taxonomical classification, four assembled bacterial genomes were identified as *Streptococcus*, two MAGs of *Lactococcus* and *Lactiplantibacillus*, single genomes belonging to *Idiomarina*, *Halomonas*, *Chromohalobacter*, *Tetragenococcus*, *Staphylococcus*, *Leuconostoc*, *Pseudomonas*, *Enterobacter*, *Lentilactobacillus*, *Enterococcus*, *Macrococcus* and *Pediococcus*.

Species assignment using ANI (average nucleotide index, 95% as reference radius) resulted in the detection of two possible new species of *Idiomarina* and *Halomonas* genera that were not assigned with a sufficient level of confidence, whereas the remaining 19 MAGs were classified up to species level (Table 3). The relative representation of individual assembled MAGs in the three metagenomes is shown in Figure S1.

Search for Bacteriocin Gene Clusters

During the search for BGC genes with antiSMASH in in silico translated MAGs obtained from three metagenomes, 86 genes from 25 gene clusters were identified, including 63 genes within 11 regions annotated as bacteriocin synthases or involved in bacteriocin synthesis (Table S3). The distribution of BGCs corresponds to the microbiological diversity: the most common clusters are class II/IIb bacteriocins of the *blp* family (13SR_bin6,

13SR_bin8, 13SR_bin10) and the lactococcin family (07SR_bin3, 13SR_bin5). For some MAGs (07SR_bin6), the complete structure of the bacteriocin synthesis cluster can be observed: ABC transporter, secretion protein, CPBP family intramembrane metalloprotease, response and transcriptional regulator, bacteriocin immunity protein and two to six subunits of bacteriocin protein [49]. In sample 07SR, the Linocin_M18 gene was found in the genome of *Levilactobacillus brevis* (07SR_bin4), the plantaricin synthesis genes were found in the *Lactiplantibacillus plantarum* genome (07SR_bin6). In samples 07SR and 13SR fragments of lanthionine-containing bacteriocin were found as Lanthipeptide Class IV clusters (07SR_bin4, 13SR_bin10). Numerous genomes (06SR_bin2, 06SR_bin3, 06SR_bin5, 07SR_bin1, 07SR_bin5) contain the YcaO domain gene that is involved in microcin antibiotics synthesis as azoline-forming protein [50].

Search for Genes of Antibiotic Resistance

Search of antibiotic resistance potential in microbiome components resulted in detection of 24 genes coding resistance-providing products in seven MAGs, as shown in Table S4. Most of the antibiotic resistance (ABR) genes contain 07SR_bin1: *oqxB* and *oqxA* encode subunits of the RND efflux pump that confers resistance to fluoroquinolone and several other antibiotics, *marA* and *crp* encode efflux pump regulator proteins in antibiotic stress (cephalosporin, glycylcycline, fluoroquinolone, macrolide, penam, tetracycline and other), *bacA* translation product confers resistance to bacitracin, *acrD* encodes aminoglycoside efflux pump, ACT-23 gene is beta-lactamase, *fosA2* gene product provides fosfomycin resistance, *msbA* codes ABC transporter which gives nitroimidazole resistance, *emrB* is a translocase gene with concomitant fluoroquinolone resistance and *baeR* encodes response regulator for aminocoumarin and aminoglycoside resistance. Only two ABR genes contained 07SR_bin2 (*lmrD* for lincosamide and *tetS* for tetracycline). Inner membrane transporter MexF which confers diaminopyrimidine, fluoroquinolone and phenicol resistance was found in 06SR_bin5. Two close genes from the MFS efflux pump family and lincosamide resistance *lmrP* and *lmrD* were detected in 13SR_bin1 and 13SR_bin5 respectively, last MAG also contains ANT(6)-la that is aminoglycoside nucleotidyltransferase. Five genes are part of 13SR_bin7 genome: *emeA* as a multidrug efflux pump gene (acridine dye resistant), *lsaA* that encodes ABC-F protein (provides resistance to lincosamide, macrolide, oxazolidinone and others), *efrA* and *efrB* encode the subunits of efflux pump (fluoroquinolone, macrolide, rifamycin resistance) and *dfrE* encodes dihydrofolate reductase that confers diaminopyrimidine resistance. Summary, in 7 out of 21 MAGs most often are genes of resistance to fluoroquinolone ($n = 9$), tetracycline ($n = 5$) and macrolide ($n = 5$) antibiotics.

Synthesis of Biogenic Amines and Some Non-Proteinogenic Amino Acids

Several bioactive amines as well as GABA can be produced by microbial communities of studied cheeses (Table S5). Genes of glutamate decarboxylase, which is responsible for GABA synthesis, were found in 06SR_bin2, four MAGs of 07SR (07SR_bin3, 07SR_bin4, 07SR_bin5 and 07SR_bin6), two MAGs of 13SR (13SR_bin3 and 13SR_bin5). Tyrosine decarboxylase which converts tyrosine to tyramine was encoded only in two MAGs (07SR_bin4, 13SR_bin7). Probably, these enzymes can also participate in beta-phenylethylamine production. No genes encoding histidine decarboxylase (responsible for histamine formation) were found. Moreover, some MAGs possessed genes of enzymes involved in putrescine formation. Putrescine production pathways in 06SR_bin2 and 13SR_bin6 include arginine decarboxylase, agmatine deaminase and N-carbamoylputrescine amidase while in 06SR_bin5 and 13SR_bin4—arginine decarboxylase and agmatine ureohydrolase. MAG 07SR_bin5 had genes encoding enzymes of both these variants. Ornithine decarboxylase genes were identified in 07SR_bin1 and 13SR_bin9.

CAZymes, Esterases and Proteases

All studied MAGs possessed CAZymes genes (glycosidases, polysaccharide lyases, carbohydrate esterases, glycosyltransferases and carbohydrate-binding modules) but their

numbers varied greatly depending on cheese sample. There were 29–49 genes in MAGs of cheese 06SR, from 66 to 111 in 07SR, and from 22 to 80 in 13SR (Table S6). Sets of glycoside hydrolases (GH) and glycosyltransferase (GT) were in the focus because these enzymes are main actors in polysaccharide decomposition and synthesis, respectively. Most universal GH found in most MAGs belonged to GH1, GH13 and GH73 (Figure 6). The first one contains enzymes with different activities including beta-galactosidases which hydrolase lactose—milk sugar. Enzymes of the GH13 family could be active against exogenous or endogenous storage alpha-glucans. In the GH73 family there is a single main activity, peptidoglycan hydrolase, so they can either in cell wall transformation during cell growth or in microbial interactions (e.g., inhibition of growth of competitors). The majority of enzymes belonging to GH2, GH8, GH25, GH65 and GH77 were encoded in MAGs obtained from 07SR and 13SR while glycosidases from GH3, GH23 and GH103 in 06SR. Among GT there were families with high abundance in all studied MAGs (GT2, GT4, GT28, GT51) but some families were sample-specific: genes of GT5 and GT35 enzymes were found only in MAGs from 07SR and 13SR, while GT9, GT19 and GT30 were found in MAGs from 06SR and 07SR.

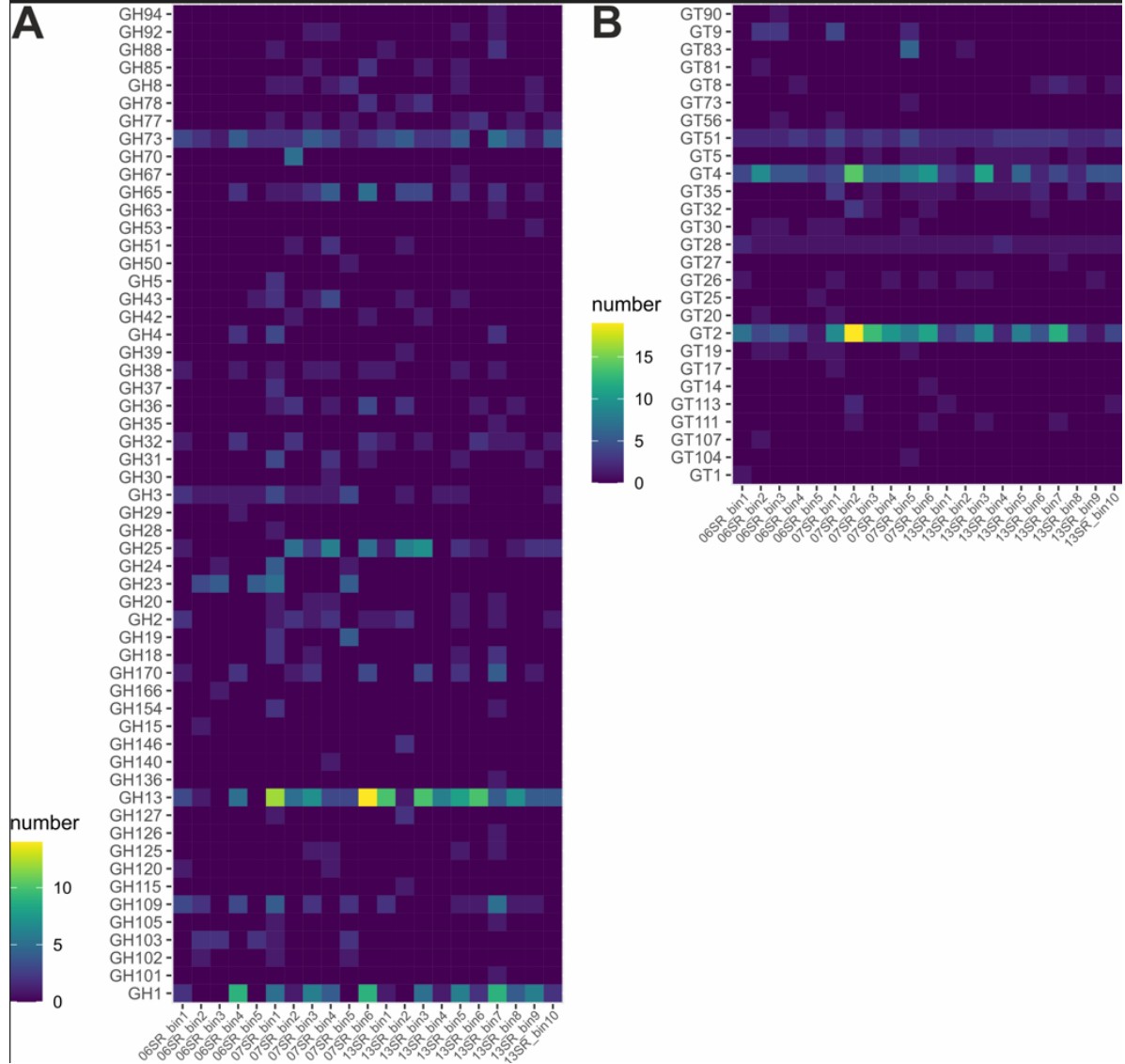

**Figure 6.** Numbers of glycoside hydrolases (**A**) and glycosyl transferases (**B**) encoded in MAGs obtained from metagenomes of three cheeses.

Genes of putative proteases were identified in all tested MAGs. Moreover, some of them have signal peptides and can be exported outside the cells or anchored at the cell surface (Table S7). Numbers of encoded extracellular proteases were 4–49, 2–41 and 1–18 in MAGs of 06SR, 07SR and 13SR, respectively. The most dominant protease families were C40, S11, S12 and M23B (found only in 06SR and 07SR).

Homologs of lipolytic enzymes affiliated to some families according to [45] were found in all studied metagenomes (Figure S2, Table S8). In total, sixteen putative esterases were encoded on 06SR: there were enzymes relatively close to Family_4 (only in 06SR_bin4), Family_6, Family_11, Family_13, Family_24, Family_32 (only in 06SR_bin5), Family_33 (only in 06SR_bin1) and Family_34. MAGs obtained from 07SR metagenome contained 24 genes of putative esterases which belong to Family_1.1 (only in 07SR_bin5), Family_1.8 (only in 07SR_bin1), Family_4, Family_6 (07SR_bin5), Family_8 (07SR_bin5), Family_10 (07SR_bin5), Family_11, Family_13, Family_15 (07SR_bin5), Family_23 (07SR_bin5), Family_29 (07SR_bin5), Family_33 and Family_34 (07SR_bin5). Twenty genes encoding esterases were found in MAGs of the 13SR metagenome. These enzymes were homologous to representatives of Family_1.10 (13SR_bin2), Family_4, Family_11 (13SR_bin3), Family_13, Family_15 (13SR_bin4), Family_31 (13SR_bin4), Family_33 and Family_35 (13SR_bin8). Some of the listed enzymes could participate in production of flavor-forming compounds.

## 4. Discussion

Fermented foods usually contain microorganisms with Generally Recognized as Safe (GRAS) status which can produce a number of beneficial by-products/metabolites. The role of probiotics in maintaining human gut health has been largely documented [1,3,51]. Artisanal dairy products may be a source of new beneficial microorganisms that are able to limit unusual inflammatory responses and metabolic abnormalities [3]. Ethnic fermented foods have traditionally been prepared by native populations and have preserved unique microbiota over the centuries. To use these distinctive microorganisms for human health improvement, it is crucial to understand the probiotic properties of the bacteria, as well as the composition of the community itself.

The history of Ossetia cheese goes back to the Scythian–Alanian times. Since ancient times, in the conditions of mountain farming, it was one of the main food products. And the recipe for making cheese has not changed much since then. The preparation of traditional Caucasian cheeses involves manual steps, such as whey draining, stomach cutting, curd stirring for salting and contact with salt, and the latter could be important microbial sources, especially halophilic bacteria [14]. Considering that examined cheeses are made from raw milk, the bacterial microbiota is expected to be highly diverse [52]. In addition, milk in different regions may contain different levels of micronutrients, proteins, lactose and fats, depending on regional conditions and the season of harvesting, all of these affect the composition of the community and their functions. Taxonomic annotation made by NGS analysis of 16S rRNA gene amplicons revealed that the microbial diversity of analyzed Caucasus cheeses did not depend on the place of origin (Figure 4B). Representatives of the genus *Lactococcus*, *Streptococcus*, *Lactobacillus* and *Lactiplantibacillus* belong to the dominant population. Although a few ASVs related to spoilage microorganisms were sporadically detected (e.g., *Brochothrix* spp., *Serratia* spp., etc.), according to the results of Illumina sequencing, no major foodborne pathogens (e.g., *Listeria monocytogenes* or *Salmonella* spp.) occurred in all analyzed samples, thus suggesting that relatively good hygiene practices had been followed during cheese production.

The dominance of specific LAB slightly correlated with the age of cheeses analyzed (in our case, with the time of ripening in salt brine) (Figure 4B). For example, *Streptococcus* and *Lactococcus* species dominated in young soft cheeses, *Lactobacillus* representatives were detected in significant amounts only in the hard cheese. The same correlation was observed for OA profile—hard cheese contained low amounts of OA, while semi-hard and soft cheeses were rich in organic acids, suggesting the intensive metabolic processes of LAB genera presented there (Figure 2). Concentrations of lactate, acetate, propionate, butyrate,

succinate and citrate were similar to values, estimated for cheeses of the same stages of ripening [53]. Lactate and acetate were the major products of fermentation followed by succinate and citrate, the latter was regarded as a substrate for succinate production by lactobacilli [54,55]. The youngest cheese in our analysis (13SR), which was produced in the farthest region, contained the relatively significant concentrations of citrate, malate and propionate, likely due to the quality of the milk used or time of ripening [56,57].

The dominance of lactococci in non-mature, and especially in young cheeses, was an expected result, according to the relevant literature [14,17,58–62]. *Lactobacillus* species, on the other hand, begin to dominate in long-ripening cheeses [14,58,63]. Most of the other LAB genera detected in this work as dominant and subdominant (*Streptococcus, Lactiplantibacillus, Lentilactobacillus, Levilactobacillus, Leuconostoc*) have often been described as a part of microbiota of spontaneously ripened cheese [14,16,60–62,64].

In genomes of *Lactococcus lactis*, detected in 07SR and 13SR cheeses, genes encoding enzymes, involved in the degradation of carbohydrates present in milk (like GH1, GH2, GH13), as well as those responsible for the synthesis of polysaccharides (GT2) that could influence the cheese texture, were found (Figure 6). In addition, lactococci actively hydrolyze milk proteins, producing flavor compounds, medium- and small-sized peptides as well as free amino acids with a high impact on flavor [65], and in our research we also identified some peptidases that could play a role in hydrolysis of milk proteins during the fermentation process. Together with *Lc. lactis*, *Lactiplantibacillus plantarum* and *Levilactobacillus brevis*, detected in 07SR cheese, are actively involved in milk sugars fermentation, due to the large variety and number of GH enzymes in their genomes. According to the literature, besides the carboxylic acid content of cheese, these LAB species also contribute to the amino acid profile [66]. The largest quantity of GTs, the main enzymes involved in polysaccharide synthesis (especially, GT2 family), was found in *Leuconostoc mesenteroides* genome from 07SR cheese, that could produce exopolysaccharides (e.g., an indigestible $\alpha$-glucan) with substantial host metabolic benefits [67].

Metagenomic analysis revealed the dominance of *Streptococcus* genera (*Streptococcus thermophilus, S. parasuis, S. macedonicus* and *S. dysgalactiae*) in the 13SR cheese microbiome. It is believed that *Streptococcus* representatives are part of the primary starters, and their source may be not only milk, but also whey and veal stomachs used for cheese making [52,68]. *S. thermophilus* causes rapid acidification of milk through the production of lactic acid, proteolytic and urease activities [69]. Our metagenome analysis also revealed a large number of genes encoding GHs in *Streptococcus* MAGs and no or poor representation of lipolytic enzymes (Figure 6 and Figure S2). It may indicate that these bacteria are not involved in fatty acid production from triglycerides, and most likely participate in the hydrolysis of milk sugars. Apparently, the lipolysis of milk fats in this cheese is responsible for *Lactobacillaceae* representatives and *Macrococcus caseolyticus* [70]. Another feature of 13SR cheese was the presence of *Lentilactobacillus kefiri,* that is commonly associated with kefir beverage, but is also detected in cheeses of different origin [61,71]. The genome of *L. kefiri* possessed GH's genes (encoding GH127, GH146 and GH39), that are practically absent in other genomes analyzed in this work and are involved in L-arabinose production (Figure 6). L-arabinose is a five-carbon sugar that is metabolized in bacteria to ethanol [72], which can affect the taste of the cheese.

LAB can produce a huge number of biologically active compounds, such as short-chain fatty acids, conjugated linoleic acid, vitamins, different peptides with antimicrobial activities, bioamines, 1,2-propanediol and much more [3]. In this work we explored the assembled genomes for the presence of genes involved in the bacteriocins and bioamines synthesis. Bacteriocins are proteins with antimicrobial activity and differ from other antibiotics in that they are all synthesized by ribosomes and that they are quite specific and often active against closely related microorganisms. Bacteriocins of lactic acid bacteria are often characterized by resistance to high temperatures and low pH but are degraded by digestive enzymes. That is why they are considered as beneficial and safe for humans and are of particular interest for applications such as food preserving agents [73]. In metagenomes

of 07SR and 13SR cheeses we found that dominant LAB may be producers of bacteriocins (lactococcin A, B, plantaricin, linocin, lanthipeptide class IV, lactobin A and lactocin F) (Table S3). Moreover, genes for the synthesis of several peptides with antimicrobial activity were found in some of their genomes (*Lc. lactis*—lactococcin A and B, *L. brevis*—linocin M18 and lanthipeptide class IV, *L. plantarum*—lactococcin A and plantaricin EF, *S. thermophilus*—lactacin F, lanthipeptide class IV and unidentified bacteriocin Bpl family class II). The identification of genes encoded linocin and lanthipeptide in the *L. brevis* genome (from 07SR sample) is particularly noteworthy, since these bacteriocins have a broad antibacterial activity (unlike lactococcins and plantaricin) and even antifungal and antiviral activity by the latter [74,75].

Biogenic amines and non-proteinogenic amino acids are often the result of the bacterial decarboxylative activity towards free amino acids in food and LAB are strong amine producers. Bioamines were of our interest because of their adverse effects on human health and the large production of these molecules specifically in cheeses. In our metagenome analysis we identified genes encoding enzymes that catalyze the production of two different bioamines (tyramine and putrescine) as well as GABA. GABA is a product of glutamate decarboxylation and is the only bioamine with a really positive effect on human health, due to its relaxing effect on muscles and its overall beneficial influence on the nervous system [76]. Tyramine is the most frequently found bioamine in cheese and acts both on the vascular level (hypertension, vasoconstriction) and on the central nervous system. It can cause not only headaches, but it can also lead to serious health consequences [77]. Putrescine is one of the most common biogenic amines found in food, with higher concentrations in fermented dairy products such as cheese and fish [78]. It is greatly responsible for the unpleasant smell of unripe foods [79]. The risk effect depends on its concentration; however, there is a lack of regulation concerning safe putrescine limits in food. We found that all the dominant bacteria in the microbiome of cheese 07SR (*Lc. lactis*, *L. brevis* and *L. plantarum*) are producers of GABA, which significantly increases the beneficial quality of this product. Also, *Pseudomonas* sp., found in this community in minor proportions, is also capable of producing this useful bioamine (Table S5). However, it also contains the putrescine synthesis genes, as does the other minor component—*Enterobacter hormaechei*, the origin of which may be related to the cow's stomach, used for cheese preparation [80]. Only two genes of glutamate decarboxylase were found in the 13SR metagenome (*L. plantarum* and *Lc. lactis*), despite the greater diversity of LAB in this community. Moreover, some representatives of *Streptococcus* genus, *Pediococcus parvulus*, *M. caseolyticus* and *Enterococcus faecalis* have been shown to be sources of putrescine and tyramine.

The latter one as well as *E. hormaechei* (from 07SR sample) possessed the highest number of antibiotic resistance-related genes in their assembled genomes among all MAGs in this work (Table S4), including ABR genes against fluoroquinolone, cephalosporin, macrolide, tetracycline etc. The detection of antibiotic resistance in dairy microflora is related to food safety issues, because the emergence and spread of antibiotic resistance is a serious problem at the present time. All this, together with the properties described above for these species, suggests against such bacteria presence in the food. As for the other components of the microbiome analyzed, the *Leu. mesenteroides* MAG (from 07SR) contained ABR genes conferring resistance to lincosamide and tetracycline, *Lc. lactis* (from 13SR) possessed genes from lincosamide resistance, and the *S. dysgalactiae* genome (also from 13SR cheese) carried genes of efflux pump belonging to the major facilitator superfamily (MFS) family of transporters that provided resistance to multiple dissimilar drugs [81]. The other LAB that made up the microbiomes of cheeses did not contain ABR genes, thus complying with one of the safety requirements for LAB used as food starters or probiotics.

The other distinctive feature was revealed during Ossetia cheese analysis. Samples 05SR and 06SR contained halophilic bacteria as the predominant majority of the community that is not commonly associated with the cheese environment [14]. Representatives of *Chromohalobacter japonicus* were prevalent in these two cheeses. *Idiomarina*, *Halomonas* and *Tetragenococcus* were detected among other dominant genera (Figure 5). These species are

commonly associated with salt fermented liquid products of vegetable origin, popular in Asian cuisine [82,83]. *Chromohalobacter* species are often detected in salty environments, including foods, not only fermented [84], and in brines, used for cheese ripening [85]. *Tetragenococcus* representatives also may dominate in cheese brines, as was demonstrated for artisanal Belgian cheese production [86]. Other halophilic (or halotolerant) genera, such as *Halomonas*, *Idiomarina*, *Marinobacter* and *Psychrobacter* constituted a significant proportion, but not a prevailing one, in some analyzed cheese samples too (08SR, 11SR and 12SR). These bacteria are often found in saline habitats, including fermented seafood [87–89], and also in sub-dominant or minor quantities in soft and semi-hard cheeses [64,90,91]. The origin of halophilic species in the cheeses studied is difficult to explain based on the results obtained in this paper. Some authors believe that raw milk microbiota could contain some halophilic bacteria [92] or salt used for brine preparation and for rubbing the cheese surface may be a source of halophiles [86,92,93]. In any case, the question remains open—why do the halophiles we discovered dominate some cheese microbiomes: are they involved in milk fermentation and cheese maturation or are they simply displacing lactic acid bacteria over time, winning competition in salty conditions?

Halophilic and halotolerant bacteria capable of lactic acid fermentation are detected and functionate in dairy products, including cheese, called HALAB, and belong to the phylum *Bacillota* [93–95]. It is known that bacteria of the *Tetragenococcus* genus, considered as HALAB, are often involved in the fermentation of sea foods as dominant microbes [96]. But all other predominant halophilic bacteria we detected in cheese samples are aerobes and belong to the phylum *Pseudomonadota* and their value in the fermentation of milk, as well as their potential probiotic properties, remain unknown.

A metagenomic analysis of the 06SR cheese was performed to clarify the role of halophilic bacteria in cheese ripening and to evaluate their possible beneficial impact. Analysis of CAZymes genes revealed that only *Tetragenococcus halophilus* and *Staphylococcus equorum* genomes possess a wide range of glycosidases involved in the hydrolysis of various sugars that may be involved in the fermentation of milk, most notably GH1 (Figure 6). The visible proportion of ASVs related to *Staphylococcus* genus in cheeses 05 and 06 is most likely due to environmental contamination. Usually, if these bacteria are found in cheeses, they are localized on the outside of cheeses (rinds) of long maturation (more than 4–6 months) [97], as in our case (Table 1). Genomes related to *Chromohalobacter*, *Halomonas* and *Idiomarina* genera contained mainly GHs genes, involved in the lysis of peptidoglycans of the bacterial cell wall (GH103, GH23, GH24, GH73). This means that these microorganisms were not involved in lactose fermentation. However, the high values of lactate and acetate in 05SR and 06SR cheeses indicate an active fermentation process, which appears to be carried out only by representatives of *T. halophilus*, streptococci (in the case of 05SR cheese) and other LAB presented in minor proportions (Figure S1). As for the synthesis of bioactive molecules by halophilic bacteria, no definite evidence was found for the ability to synthesize bacteriocins by any of the components. Only genes encoding proteins potentially involved in microcin synthesis (Table S3) have been found in the genomes of *Chromohalobacter*, *Halomonas* and *Idiomarina* genera, but their antibacterial capabilities remain enigmatic [98]. At the same time, putrescine genes were detected in three genomes from the 06SR cheese sample (*Staphylococcus equorum*, *C. japonicus* and *Idiomarina* sp.), and GABA gene only in *C. japonicus* MAG. All of the above points against the beneficial effects of the dominance of halophiles in cheeses (except of HALAB strains of *T. halophilus*, that can be used as a health-promoting probiotic [99,100]), and the use of such products in food is rather undesirable.

## 5. Conclusions

Artisanal Caucasus cheeses coagulated with animal rennet represent a source of still-undiscovered microbial diversity, and the present study contributes to increase the knowledge of the microbial species naturally occurring in dairy products. In this study, in addition to lactic acid bacteria species that contribute to fermentation and ripening,

other unexpected species of halophilic bacteria were also found and even dominated in some of them. Metagenome analysis revealed that halophiles (except for *T. halophilus*) were not involved in milk fermentation and they most likely cannot be considered as probiotic because they do not contain the genes responsible for the synthesis of beneficial products/metabolites, but instead they may be involved in the synthesis of inappropriate ones, such as bioamines. In contrast, analysis of the genomes of most LAB revealed the presence of genes encoding beneficial properties for humans. This also includes the absence of antibiotic resistance genes in most LAB, detected in analyzed cheeses. Thus, the use of NGS technologies makes it possible to relatively quickly assess the microbial composition and functionality of the fermented dairy products, which can contribute to the targeted exploration of selected strains with specific probiotic properties, isolated from the products.

**Supplementary Materials:** The following supporting information is available online at https://www.mdpi.com/article/10.3390/fermentation9080719/s1, Table S1. Accession numbers of SRA used in this work. Table S2. Relative abundance of 182 taxa identified in 288 amplicon sequence variants. Table S3. Bacteriocin gene clusters. Table S4. Genes of antimicrobial resistance found in MAGs. Table S5. Enzymes involved in biogenic amines and GABA biosynthesis. Table S6. CAZymes encoded in MAGs. Table S7. Extracellular proteases encoded in MAGs. Table S8. Esterases genes found in MAGs obtained from metagenomes of three cheeses (06SR, 07SR, 13SR). Figure S1. Relative abundance of MAGs in metagenome samples. As a measurement of relative abundance here used normalized GCPM (gene copies per million reads), obtained by estimation of the abundance of individual bins and compiling average values for samples. Figure S2. Putative esterases encoded in MAGs obtained from three cheese metagenomes (06SR, 07SR and 13SR).

**Author Contributions:** Sampling, A.I.S. and G.S.K.; sample processing for OA analysis, K.S.Z., L.A.G. and P.A.S.; HPLC OA analysis, K.S.Z.; DNA isolation and sequencing, A.A.K., P.A.S. and K.S.Z.; bioinformatic analysis, I.P.G., I.V.K. and A.G.E.; writing—original draft preparation, T.V.K., K.S.Z., I.P.G. and A.G.E.; figures preparation, A.G.E., K.S.Z. and I.P.G.; writing—review and editing, T.V.K., I.P.G., A.A.K. and A.G.E.; supervision, T.V.K. All authors have read and agreed to the published version of the manuscript.

**Funding:** This work was supported by the Ministry of Science and Higher Education of the Russian Federation in the framework of the Federal scientific–technical programme of the genetic technologies development for 2019–2027 (Agreement No. 075-15-2021-1401, 3 November 2021).

**Institutional Review Board Statement:** Not applicable.

**Informed Consent Statement:** Not applicable.

**Data Availability Statement:** https://www.ncbi.nlm.nih.gov/bioproject/789261, accessed on 23 June 2023; https://www.ncbi.nlm.nih.gov/nuccore/JARIFJ000000000, accessed on 23 June 2023; https://www.ncbi.nlm.nih.gov/nuccore/JARIFN000000000, accessed on 23 June 2023; https://www.ncbi.nlm.nih.gov/nuccore/JARIFP000000000, accessed on 23 June 2023.

**Conflicts of Interest:** The authors declare no conflict of interest. The funders had no role in the design of the study; in the collection, analyses, or interpretation of data; in the writing of the manuscript, or in the decision to publish the result.

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
