# Peer review of "The Bacterial Microbiota of Artisanal Cheeses from the Northern Caucasus"

_fermentation, doi:10.3390/fermentation9080719_

Round 1

Reviewer 1 Report

This study presents culture-independent approach using 16SrRNA gene amplicons sequencing and metagenomics of selected cheeses for determination of microbiota and their functional and safety characteristics. The work was enriched in the organic acids profile analyses of cheeses.

I generally find this article very interesting, with well-applied methods and  cleary presented results, however the main weak point is the lack of biological repetitions or at least analysis of the same samples in time. Usually one of such analytical strategies is required (or at least good practice). As I found similar publications in Fermentetion journal (https://doi.org/10.3390/fermentation7030174) I leave this decission to the Editor and not reject based on this factor.

Minor comments:

The current taxonomy of Procaryota can be found at: https://lpsn.dsmz.de/ . Please, give within the manuscript correct names of taxa (Bacillota, Pseudomonadota) giving in brackets the former names or synonims. Add correct names also to the Abstract.

Give the full name of the species when you write it for the first time, then use shortcuts (eg L. plantarum).

Decide whether you use one-letter shortcuts for genus component or more letters. Unify with the manuscript all names (eg. Lactococcus - L. or Lc.; Lactobacillus - L. or Lb).

Minor comments:

Line 54: correct the sentence: "In ancient Greece, cheese making was as well known as we are". Should that be: "In ancient Greece, cheese making was as well known as in our ages"?

Lines 186 and 199: use subscript for exponents: "1e10-10", "1e10-5"

Minor comments:

Line 26: correct to "cheeses were dominated"

Lines 29-30: correct grammar in the sentence: "Halophilic lactic acid bacteria belonging to the Tetragenococcus genus consisted of 7.9-18.6% in 05SR and 06SR microbiomes.

Line 75: correct to "[7] etc."

Author Response

Dear Reviewer.

We thank you for your appreciation of our work. We are very glad that you found it interesting. Your remark about the lack of biological repetitions (or in time) is absolutely on point. The importance of statistical strategies in biological studies is undoubted. However, it is very difficult to realize in our work due to the specificity of artisanal dairy food. The process of making the cheeses under study is manual and is influenced by a huge number of factors: the amount of rainfall that has fallen this year, the ratio and composition of the herbs that the cow has eaten the humidity of the room, etc. The villagers who make these cheeses themselves say that the taste/color of cheese can vary, even if they were made yesterday or today by one farmer. Therefore, an adequate statistically reliable study of the microbiomes of farm dairy products is only possible over a very long period of time (several years). As for the analysis of the same samples in time, this is also complicated due to hard-to-reach of many mountain villages in the North Ossetia. This paper is a first step towards understanding the composition of the communities of national products produced in the Caucasus and their nutritional value.

Minor comments

The current taxonomy of Procaryota can be found at: https://lpsn.dsmz.de/ . Please, give within the manuscript correct names of taxa (BacillotaPseudomonadota) giving in brackets the former names or synonims. Add correct names also to the Abstract.

Surely, you are right. Since the taxonomic assignment of ASV in our research was performed using Silva 138.1 database [doi:10.1093/nar/gks1219] (and it differs somewhere from https://lpsn.dsmz.de/), in the first version of the Manuscript we had decided to mention the old names in the text of the paper, in order not to confuse the reader. In the text, however, we had noted the currently correct names (section 3.2.1.). But since that is not enough, we accept your comment and for this purpose we have corrected the old names of taxa to the relevant ones (with mention of synonyms in brackets) in the whole text and figure 3.

Give the full name of the species when you write it for the first time, then use shortcuts (eg L. plantarum).

Thank you. We have corrected when appropriate.

Decide whether you use one-letter shortcuts for genus component or more letters. Unify with the manuscript all names (eg. Lactococcus - L. or Lc.; Lactobacillus - L. or Lb).

Done.

Line 54: correct the sentence: "In ancient Greece, cheese making was as well known as we are". Should that be: "In ancient Greece, cheese making was as well known as in our ages"?

Done.

Lines 186 and 199: use subscript for exponents: "1e10-10", "1e10-5"
Done.

Line 26: correct to "cheeses were dominated"

Done.

Lines 29-30: correct grammar in the sentence: "Halophilic lactic acid bacteria belonging to the Tetragenococcus genus consisted of 7.9-18.6% in 05SR and 06SR microbiomes.
We changed the verb “consisted of” to “accounted for”. Hopefully, it’s better this way.

Line 75: correct to "[7] etc."

Done.

Reviewer 2 Report

The topic of the manuscript: „The bacterial microbiota of artisanal cheeses from the Northern Caucasus falls within the thematic scope of FERMENTATION.

The aim of the study was to determine the diversity of the microbiome of 12 traditional samples of home-made authentic cheeses from the Caucasus region, as well as to investigate the gene sets associated with metabolic activity and probiotic functions that may be involved in sensory characteristics, safety and beneficial properties. The study used culture-independent NGS analysis based on 16S rRNA gene amplicons of bacterial communities of cheese samples and metagenomics of 3 of them.

This is a very interesting topic and worth discussing. I consider the results of halophilic and halotolerant bacteria present in the tested cheeses to be the most interesting and new element of the work. I believe that these results may be an interesting contribution to further research in other countries on this group of microorganisms in other types of cheese. I think that an interesting aspect would be to analyze the results in terms of psychrotrophic properties of both LAB and others (Pseudomonas).

In general, I rate the manuscript very highly, which is why I suggest the Authors a few minor corrections that will only improve their article.

All comments were introduced in the review mode to the attached pdf file.

Author Response

Dear Reviewer.

We thank you for your appreciation of our work. We are very glad that you found it interesting. We corrected all remarks that you indicated. As for the Table 1, that it is missed in Materials and Methods, we partly agree with you, since it contains the information about samples characterization. But it also contains the results obtained in this work, so we decided to place it just in the beginning of the Results section. We referenced the Table 1 in the section 2.1 in order to reduce the confusion. Also we redesigned the Picture 1, tried to make it more visible, and made the captions more noticeable. Thank you for your comment.